# Benford's Curse: Tracing Digit Bias to Numerical Hallucination in LLMs

**Jiandong Shao**[1]*   **Yao Lu**[2],   **Jianfei Yang**[1]†
[1]Nanyang Technological University,   [2]University College London
shamyshao@gmail.com,   yao.lu@cs.ucl.ac.uk,
jianfei.yang@ntu.edu.sg

## Abstract

Large Language Models (LLMs) exhibit impressive performance on complex reasoning tasks, yet they frequently fail on basic numerical problems, producing incorrect outputs. Inspired by Benford's Law, a statistical pattern in which lower digits occur more frequently as leading digits, we hypothesize that the skewed digit distributions in web-collected corpora may be learned by LLMs during pretraining, leading to biased numerical generation. To investigate the hypothesis, we first examine whether digits frequencies in pretraining corpus (OLMo2) follows Benford's law. We then construct an evaluation benchmark in which the ground-truth digits are uniformly distributed within each of the seven numerical reasoning tasks. Our evaluation results demonstrate that leading open-source LLMs show a consistent pattern of digit bias that resembles Benford's law. Through logit-lens tracing and neuron-level dissection, we identify that this bias arises predominantly from a small subset of highly digit-selective feed-forward network (FFN) neurons in the deeper layers. Finally, we demonstrate that pruning these neurons mitigates imbalanced overgeneration and partially corrects erroneous outputs, providing causal evidence that fine-grained pretraining digit bias can propagate into model behavior. Our findings reveal a fundamental connection between corpus-level statistics and symbolic failure modes in LLMs, offering a new lens for diagnosing and mitigating hallucinations in numerical tasks. Code and data is made available here: `https://github.com/shamy28/Benford-Curse`.

## 1 Introduction

Large Language Models (LLMs) have demonstrated impressive performance not only on daily reasoning tasks but also on challenging domains such as mathematical competition problems, showcasing strong symbolic and logical capabilities [1, 2, 3, 4, 5, 6, 7]. Yet, despite these amazing success, LLMs frequently fail on surprisingly simple numerical reasoning—often producing mathematically incorrect or logically inconsistent outputs [8, 9, 10]. This discrepancy prompts a fundamental and pressing question: why are LLMs so prone to hallucinations when dealing with basic numbers?

Previous studies [11, 12, 13, 14] have identified duplication bias in pretraining corpus as a major contributor to hallucinations in natural language tasks. For instance, when prompted with "list some red fruits, excluding apples", LLMs frequently still mention "red apples" likely due to the high frequency of co-occurring phrases such as "red apples, watermelon, cherries, and strawberries" in the training data. This over-reliance on memorized patterns causes the model to overlook explicit instructions in the prompt [11]. Given this strong pattern-completion tendency in language, a natural

---

*This work was conducted during the author's research internship at MARS Lab, NTU.
†Corresponding author.

> **Task: Identifying the Last Term in a Sequence**
>
> **Prompt 1:** What is the result of multiplying the sequence's last term by two? [..., 2.95, 1.81, 8.26]
> **Model Generation:** To find the result of multiplying the sequence's last term by two, we need to identify the last term in the sequence and then multiply it by 2.
> The last term in the sequence is **1.81 ✗**
> *(Incorrect: the model mistakenly identifies the second-to-last number as the last term.)*
>
> **Prompt 2:** What is the result of multiplying the sequence's last term by two? [..., 7.43, 8.51, 1.05]
> **Model Generation:** To find the result of multiplying the sequence's last term by two, we need to identify the last term in the sequence and then multiply it by 2.
> The last term in the sequence is **1.05 ✓**
> *(Correct: the model correctly identifies the last number in the sequence.)*

Figure 1: Illustration of digit bias in identifying the last term of a numerical sequence. LLaMA3.1-8B-Instruct is asked to multiply the final term of a sequence by two, but first must identify the last item. In the first case, the model incorrectly selects a smaller intermediate value (1.81) instead of the actual final term (8.26). In the second, it correctly selects 1.05. This asymmetry suggests a bias toward smaller digits when the model is uncertain, revealing a subtle form of numerical hallucination. Detailed accuracy comparisons are shown in Figure 2.

question arises: might similar memorization-driven biases emerge in numerical reasoning tasks, potentially leading to systematic numerical hallucinations?

Benford's Law [15], a striking statistical phenomenon, was first observed in 1881 by astronomer Simon Newcomb, who noticed that the earlier pages of logarithmic tables—those starting with digit 1—were more worn than others [16]. This empirical observation was later formalized by physicist Frank Benford in 1938, who verified the pattern across over 20 diverse datasets [17]. Benford's Law states that the probability that a number begins with digit $d \in \{1, \ldots, 9\}$ is given by $P(d) = \log_{10}(1 + \frac{1}{d})$. This implies that digit 1 occurs as the leading digit in about 30% of cases, while digit 9 appears less than 5%. This law has since been observed across various domains such as economic records, scientific measurements, and population statistics, and thus naturally emerges in human-written text [15, 18, 19]. Given this ubiquity, it is natural to ask whether the pretraining corpora of LLMs, which are largely composed of web-scraped data, also reflect this skewed digit distribution. If so, could LLMs internalize such patterns during training, and might this lead to a form of *digit bias* that contributes to *numerical hallucinations*, even on otherwise simple numerical reasoning tasks?

To investigate this question, we begin by examining the digit distribution in the Olmo-mix-1124 pretraining corpus [20] and confirm a strong digit skew consistent with Benford's Law. To test whether this corpus-level skew propagates into model behavior, we construct a diagnostic benchmark with uniformly distributed ground-truth digits, eliminating task-induced priors. Empirical results show that LLMs consistently overgenerate smaller digits, despite uniform targets. Further analysis reveals that when the model generates incorrect answers, the distribution of first error digits(e.g., for ground truth 758 and prediction 714, the first error digit is 1) exhibits an even stronger skew toward smaller values. To better understand the underlying mechanisms, we analyse the model's internal representations. First, we apply the Logit Lens [21] to trace how digit preferences evolve across layers and find that the preference for smaller digits often emerges strongly in the later layers. Second, we disentangle the contributions of FFN and self-attention, finding that the bias is primarily driven by the FFN. Third, by quantifying the digit selectivity of FFN neurons, we find that their aggregated preferences form a skewed distribution favouring smaller digits, closely mirroring the statistics of the pretraining corpus. Motivated by these insights, we explore a lightweight neuron-level intervention that prunes a small set of biased neurons. This intervention can partially mitigate overgeneration and correct certain hallucinated outputs, offering further evidence of the causal role digit bias plays in numerical hallucination.

The contributions of this work are summarized as follows: (1) We formulate and investigate a fundamental question chain: does the skewed digit distribution in LLMs' pretraining corpus induce digit generation bias, leading to numerical hallucination? (2) To explore this, we construct a diagnostic benchmark with uniformly distributed target digits and observe that LLMs consistently overgenerate smaller digits. Notably, the first error digits exhibit an even stronger skewed distribution, suggesting a

bias-driven mechanism behind numerical hallucination. (3) Using logit lens tracing and neuron-level analysis, we identify that this digit bias primarily originates from a subset of highly digit-selective feedforward (FFN) neurons, particularly concentrated in the later layers of the model. (4) Finally, we show that pruning these neurons can partially mitigate hallucination, offering causal evidence that digit bias is a contributing factor to numerical hallucination. By identifying a fine-grained statistical artifact as a mechanistic failure point, our work highlights a critical yet underexplored source of errors in numerical reasoning and underscores the importance of addressing pretraining-induced biases to develop more reliable language models.

## 2   Related Works

**Numerical Hallucinations in LLMs.**   While there is no universally accepted definition of numerical hallucination, the term broadly refers to a language model's tendency to generate numerically incorrect, inconsistent, or implausible outputs, despite syntactically or semantically coherent completions. These include phenomena such as miscounting, incorrect number reproduction, inappropriate number substitution, or generating fabricated values. Prior work has largely attributed these numerical hallucinations to reasoning deficiencies [22], suboptimal tokenization schemes [23], and misalignment [24]. Razeghi et al. [25] further showed that model accuracy in numerical tasks correlates with token frequency in pretraining data, suggesting a statistical basis for some failures. However, these studies do not examine how digit-level statistical patterns manifest during generation. We highlight this as a distinct and measurable form of numerical hallucination overlooked in prior work.

**Dataset Bias as a Source of Hallucination.**   Prior studies have increasingly identified dataset bias as a key driver of hallucination in LLMs [14, 11, 13, 25, 12], suggesting that many hallucinations stem not from architectural flaws but from imbalances in the pretraining corpus. For example, overrepresented entities or linguistic patterns in large-scale web data can lead models to repeat them even in inappropriate contexts [11]. However, existing work [26, 27, 28] has primarily focused on semantic-level biases, such as spurious facts or misattributions, while overlooking lower-level statistical regularities. In particular, the extent to which digit-level statistical pattern in the training data influences the model's numerical generation remains largely unexplored. Our work fills this gap by showing that frequency pattern over digits in the pretraining corpus is systematically reflected in model generations, providing a concrete statistical basis for a subset of numerical hallucinations.

## 3   Investigating Digit Bias in LLMs

Benford's Law describes a logarithmic distribution over leading digits in naturally occurring numerical data, where smaller digits appear with higher frequency than larger ones. This prompts a critical question: does the pretraining corpus, largely sourced from real-world Internet text, also exhibit a similar skewed digit distribution? If so, how might such a skewed distribution influence digit generation in LLMs? To motivate this investigation, we begin with a simple diagnostic experiment illustrated in Figure 1. In this basic sequence identification task, the model demonstrates noticeably higher accuracy when the final digit to be predicted is small (1 or 2) compared to when it is large (8 or 9), as shown in Figure 2. This asymmetry suggests a preference for generating small digits, hinting at a potential internal digit bias that could underlie numerical hallucinations.

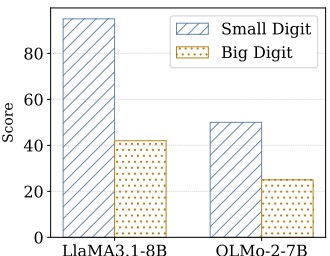

Figure 2: Accuracy in the sequence last term identification task shown in Figure 1. Digits in the range 1-3 (low-value) are recognized with significantly higher accuracy compared to digits in the range 8-9 (high-value), indicating a generation bias toward smaller digits.

**Digit Distribution in Pretraining Corpus.**   To empirically assess whether the digit distribution in pretraining corpus aligns with Benford's Law, we analyze the Olmo-Mix-1124 corpus [20], a widely-used 22.4TB data collection curated for training open-source LLMs. As shown in Figure 4a, the digit frequencies strongly align with Benford's Law, exhibiting a pronounced logarithmic distribution that

Table 1: Examples of tasks in the Digit Bias Benchmark, Covering seven numerical reasoning tasks.

| Task Category | Examples |
|---|---|
| Add or Sub | 1. What is -0.9121789 minus -6? |
| | 2. Add 4292 and 597069553.5. |
| Multiplication | 1. Multiply 13862 and 0.5. |
| | 2. What is -464 times -3? |
| Division | 1. Solve 81190 divided by 2. |
| | 2. Calculate the division of 40380 by 5. |
| Evaluate | 1. Let c(a) = -2*a - 10. What is c(-1)? |
| | 2. Let l(f) = f**3 - 3*f**2 + f + 3. Give l(2). |
| Nearest Integer Root | 1. What is the fifth root of 29889504 to the nearest integer? |
| | 2. What is 4528941 to the power of 1/9, to the nearest integer? |
| Linear_1d | 1. Solve 33*r = 8*r - 137 + 712 for r. |
| | 2. Solve 309*j + 23940 = -356*j for j. |
| Sequence Next Term | 1. What comes next: 532118, 1064232, 1596346? |
| | 2. What is next in 704, 506, 334, 188, 68? |

favors smaller digits. These findings suggest that LLMs are likely to encounter digit distributions that are heavily biased toward smaller values during training.

**A Benchmark for Measuring Digit Bias.**   To investigate whether skewed digit distribution in pretraining data leads to generation bias, we introduce the **Digit Bias Benchmark**, a suite of seven numerical reasoning tasks designed to yield uniformly distributed ground-truth digits. Tasks include *add or sub*, *multiplication*, *division*, *evaluate*, *nearest integer root*, *linear_1d*, and *sequence next term*, primarily adapted from DeepMind's Mathematics Dataset [29]. Each task contains over 1,000 examples, with answer sets carefully constructed to ensure uniform digit distribution: when pooling all digits from all positions across all answers within a task (e.g., the answer "132" contributes three digits: 1, 3, and 2), each digit 0-9 appears approximately 10% of the time. This design enables us to disentangle generation bias from task-induced digit distribution effects, allowing a more controlled evaluation of digit-level preferences in LLMs. Representative examples for each task are listed in Table 1.

**Empirical Evidence of Digit Bias.**   We evaluate six open-source LLMs on the proposed Digit Bias Benchmark, including models from LLaMA [30], Qwen [31], Gemma [32], OLMo [33] and Mistral [34] families. By design, the benchmark enforces a uniform digit distribution in its ground truth, so any deviation in the model's output distribution directly reflects inherent generation bias. As shown in Figure 3a, Mistral-7B exhibits a strong and consistent over-generation of smaller digits. For example, digit 1 often appears over 12% of the time, while digits such as 8 and 9 are severely underrepresented. This trend closely parallels the skew found in the pretraining corpus, reinforcing the hypothesis that the bias originates from corpus-level statistics. To further probe the behavioral

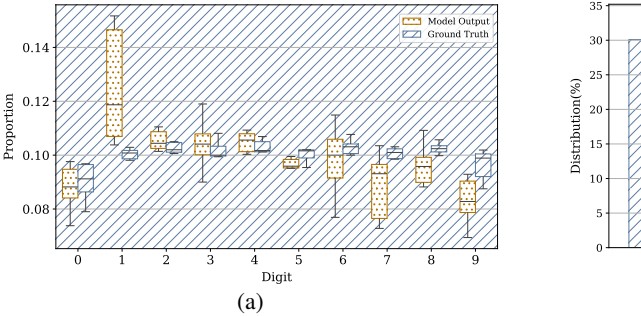
(a)

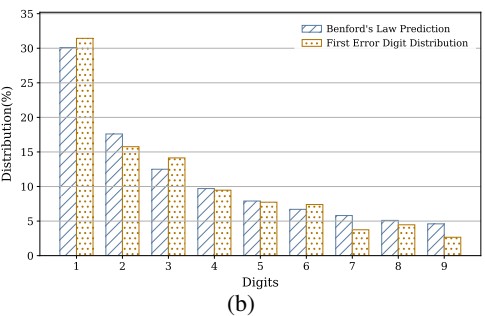
(b)

Figure 3: Digit bias observed in the Digit Bias Benchmark with Mistral-7B. (a) The boxplot shows the distribution of digits in generated answers across all tasks, revealing a significant overrepresentation of smaller digits despite the benchmark's uniform ground-truth distribution. (b) The distribution of digits at the first error position exhibits an even stronger skew toward smaller values, closely following Benford's Law. Together, these results suggest that digit bias shapes not just preferences but also the numerical hallucination.

impact of this bias, we conduct a fine-grained error analysis. For each incorrect output answer, we identify the first digit where the model's output diverges from the ground truth and record its value (e.g., for ground truth 758 and prediction 714, the first error digit is 1). As shown in Figure 3b, the distribution of these "first error digits" exhibits an even stronger skew toward smaller values, closely following Benford's Law. This finding suggests that digit bias not only affects overall preferences but also distorts the model's generation trajectory when it deviates from the correct answer, implicating it as a potential driver of numerical hallucination. More results are shown in Figure 9.

## 4 Analyzing the Mechanisms of Digit Bias

### 4.1 Probing Layerwise Behavior with Logit Lens

**Method: Logit Lens.** The LLMs used in this paper are based on a decoder-only architecture composed of stacked Transformer layers [35] with residual connections [36]. During a standard forward pass, the hidden representation $h_n^{(0)} \in \mathbb{R}^d$ for the final input token $x_n$ is retrieved from a learned embedding table. This representation is then iteratively updated through each Transformer block, incorporating outputs from both self-attention and feed-forward submodules via residual addition. After the final layer, the hidden state $h_n^{(L)}$ (with $L$ denoting the total number of layers) is normalized and projected via the unembedding matrix $U \in \mathbb{R}^{v \times d}$, yielding logits over the vocabulary of size $v$. Since hidden states across all layers share the same dimensionality, intermediate representations can also be projected via $U$ to obtain layer-wise token distributions. This technique, known as the *logit lens* [21], allows us to visualize how the model's token predictions evolve across layers, providing interpretable insights into the generation process. To explore the origins of digit bias, we apply the logit lens to trace layer-wise changes in predicted digits.

**Datasets.** As observed in Section 3, LLMs consistently tend to favor smaller digits in numerical reasoning tasks. This suggests that, under conditions of uncertainty or hallucination, the model is more likely to generate smaller digits. Therefore, to trace the internal origins of this bias, hallucinated outputs should be the focus. However, hallucinations in multi-step reasoning tasks are often hard to localize, as the precise point of failure is difficult to identify. To simplify this analysis, we begin with single-step reasoning tasks (basic arithmetic tasks from the benchmark), where hallucination can be precisely localized by identifying the first incorrectly generated digit. Then, to expand this analysis beyond a limited set of hand-selected cases, we develop an entropy-based automated sampling strategy grounded in observations from these hand-analyzed examples to identify uncertain samples. Specifically, we compute the entropy of the model's output digit token distribution at each layer. Samples with entropy exceeding a threshold (e.g., >3.0 at layer 26) are flagged as potentially biased. Applying this criterion to the *Evaluate* task yields a larger set of high-uncertainty samples, enabling a more systematic investigation of digit bias within the model's internal dynamics.

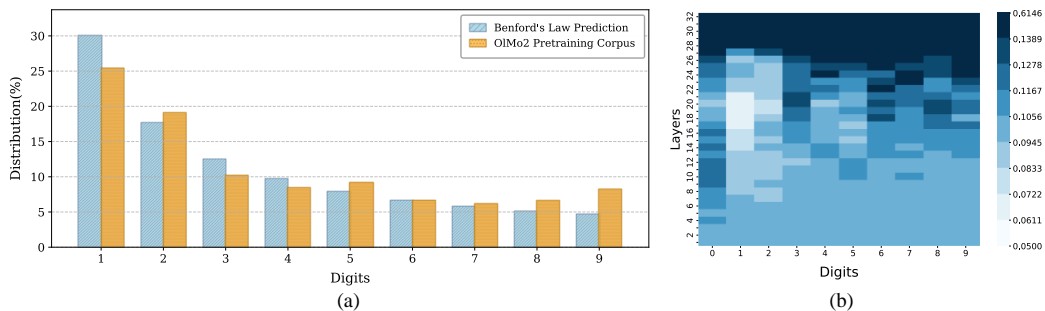

Figure 4: (a) The histogram compares the digit distribution predicted by Benford's Law with that of the OlMo-mix-1124 corpus, showing their degree of similarity. (b) The heatmap of digit probabilities across layers obtained via Logit Lens. Smaller digits remain relatively undistinguished in early and middle layers, whereas larger digits show sharper activations earlier. This indicates that the overgeneration of small digits is driven by preferences formed in the final layers.

**Layerwise Visualization of Digit Bias.** By applying the logit lens [21] to high-uncertainty samples that ultimately generate different digits, we obtain the layer-wise digit preference heatmap shown in Figure 4b. The visualization reveals a consistent pattern: under similar levels of uncertainty, smaller digits tend to exhibit stronger generation signals only in the later layers, whereas larger digits show earlier and more gradual emergence. For instance, digit 1 typically shows little preference in intermediate layers but becomes strongly favored in the final few. These results indicate that digit bias is not uniformly distributed across the network, but instead predominantly arises in the final layers.

**Layerwise Bias Trends.** To gain deeper insight into how digit bias emerges in the later layers, we conduct a layer-wise analysis focusing on how the model's digit preferences evolve across later layers. Specifically, we examine samples where digit token probabilities are approximately equal at an intermediate layer and then trace how these probabilities shift in subsequent layers. As shown in Figure 5, the model begins to exhibit a clear preference for smaller digits in later layers, with their probabilities increasing more sharply compared to larger digits. This further supports our observation that digit bias is primarily concentrated and amplified in the final layers of token generation.

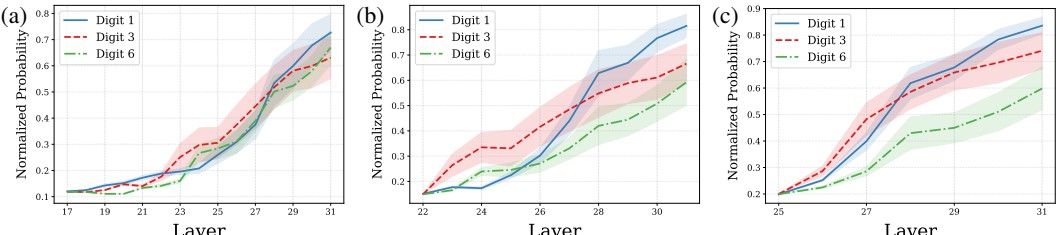

Figure 5: **Digit probability trajectories in later layers.** Starting from layers where digits 1, 3, and 6 have equal probabilities (layers 17, 22, and 25 respectively), we trace their subsequent evolution. Digit 1 consistently gains probability more rapidly than the others, suggesting that the model amplifies its bias toward smaller digits during final token prediction stages.

## 4.2 Digit Selectivity Score

To quantify how strongly a hidden vector favors a specific digit more precisely, we propose the **Digit Selectivity Score (DSC)**. Given a hidden vector $h^d$, we first project it into vocabulary logits via the model's unembedding matrix. For each digit token $i \in \{0, \ldots, 9\}$, we compute its rank within the full vocabulary. Let $S$ denote the sum of ranks of all digit tokens. Then, the DSC for digit $i$ is defined as $\text{DSC}_i = S/\text{rank}(i)$. Higher values indicate stronger selectivity for that digit. While normalized probabilities can indicate digit preference to some degree, they often fail to provide meaningful differentiation when all digit probabilities are low, leading to misleading or noisy interpretations. In contrast, DSC offers a more sensitive and robust measurement of digit selectivity for any vector in the model's hidden space.

## 4.3 Self-Attention vs. FFN

Our goal in this section is to determine whether digit bias primarily originates from the self-attention, the FFN, or their combined effect. Prior work [37, 38, 39] using causal mediation analysis has shown that, in arithmetic tasks, digit generation is predominantly driven by the FFN, while only a small number of attention heads contribute by storing simple arithmetic facts or attending to operands and operators. To examine whether digit bias follows a similar pattern, we compare the correlation between the DSCs of each layer's residual stream and the DSCs of its self-attention and FFN. Let $\mathbf{d} = (d^0, d^1, \ldots, d^L)$ denote the DSCs of the residual stream across all layers, $\mathbf{d}_{\text{ffn}} = (d^0_{\text{ffn}}, d^1_{\text{ffn}}, \ldots, d^L_{\text{ffn}})$ the DSCs computed from FFN outputs, and $\mathbf{d}_{\text{attn}} = (d^0_{\text{attn}}, d^1_{\text{attn}}, \ldots, d^L_{\text{attn}})$ those from self-attention outputs. We then compute the Spearman correlation between $\mathbf{d}$ and each of $\mathbf{d}_{\text{ffn}}$ and $\mathbf{d}_{\text{attn}}$ to assess which component more closely aligns with the overall digit bias in the model's hidden representations.

**Comparison Results.** Figure 6 presents the Spearman correlation results across layers. In intermediate layers, the residual DSC is strongly correlated with the self-attention output, while in the later layers, the residual DSC shows a much stronger correlation with the FFN output and a very weak

correlation with the self-attention output. Given that digit bias emerges most prominently in later layers, this suggests that the FFN module in the later layers is the main contributor to such bias.

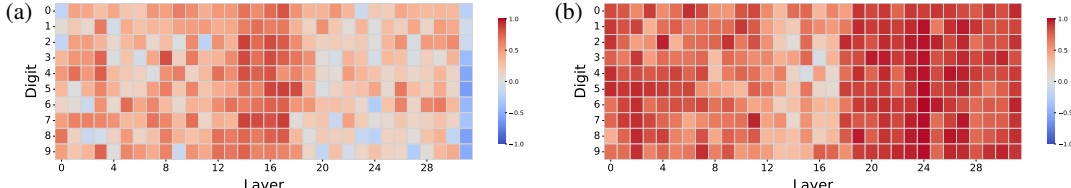

Figure 6: (a) Self-attention DSC vs. residual DSC. (b) FFN DSC vs. residual DSC. Self-attention exhibits moderate correlation in middle layers, while the FFN shows a stronger correlation in the later layers, highlighting the FFN's role in driving digit bias.

### 4.4 FFN-level Analysis

Having established that digit bias primarily emerges from the FFN in later layers, we now investigate why FFNs in these layers tend to induce such bias.

**Interpreting the FFN.** The feed-forward network (FFN) at layer $m$ produces its output as:

$$\mathbf{f}_{\text{out}}^{(m)} = \text{FFN}_{\text{in}}(\mathbf{f}_{\text{in}}^{(m)}) \cdot \mathbf{W}_{\text{out}}^{(m)} = \mathbf{f}_{\text{int}}^{(m)} \cdot \mathbf{W}_{\text{out}}^{(m)} = \sum_{n=1}^{d_{\text{int}}} f_{\text{int}}^{m,n} \cdot \mathbf{w}_{\text{out}}^{m,n} \tag{1}$$

Here, $\mathbf{f}_{\text{in}}^{(m)} \in \mathbb{R}^d$ and $\mathbf{f}_{\text{out}}^{(m)} \in \mathbb{R}^d$ are the input and output vectors of the FFN at layer $m$, $\mathbf{f}_{\text{int}}^{(m)} \in \mathbb{R}^{d_{\text{int}}}$ is the post-activation intermediate vector, and $\mathbf{W}_{\text{out}}^{(m)} \in \mathbb{R}^{d_{\text{int}} \times d}$ is the down-projection matrix. The pair $(f_{\text{int}}^{m,n}, \mathbf{w}_{\text{out}}^{m,n})$ represents the activation and output direction of the $n$-th neuron in the layer. This formulation aligns with the view of Geva et al. [40], who interpret each FFN neuron as a "key-value memory" unit, where the output direction $\mathbf{w}_{\text{out}}^{m,n}$ encodes semantic information. When projected through the unembedding matrix $U$, it reveals a token preference distribution that can be interpreted as the neuron's functional role.

**What Drives Digit Bias in the FFN?** The output of an FFN layer is shaped not only by the input representation but also by the directional structure of its learned parameters. While inputs are often complex and difficult to interpret, the output directions of individual neurons offer a more interpretable handle. Building on the interpretability analysis above, we examine whether the skewed digit distribution from the pretraining corpus is reflected in the FFN's parameters. To this end, we compute the DSC of each neuron with respect to each digit, and then aggregate these scores to obtain a model-wide selectivity profile. We find a strong correlation between this profile and the digit frequency distribution in the pretraining data ($r = 0.949$, Pearson), suggesting that the FFN not only encodes general numerical knowledge but also internalizes corpus-level frequency biases. Figure 7 shows the DSC distributions of the top 1000 most selective neurons for digit 1 and digit 7. Neurons associated with digit 1 consistently exhibit higher selectivity scores than those for digit 7, indicating that the model dedicates a larger portion of its representational space to more frequent digits. This uneven allocation likely contributes directly to the emergence of digit bias in generation.

## 5 Debiasing Method: Probing the Causal Role of Digit Bias

Our analysis shows that LLMs not only overgenerate smaller digits but also exhibit a stronger tendency to generate them when first deviating from the correct answer, following a pattern consistent with Benford's Law. This suggests that digit bias may shape not just generation preferences but also the trajectory of numerical hallucination. However, statistical alignment alone does not establish whether digit bias plays a direct causal role in numerical hallucination. To probe this link, we introduce a lightweight neuron pruning intervention. Rather than aiming to improve overall accuracy, this approach tests whether selectively suppressing neurons most biased toward digit 1 can reliably correct erroneous outputs.

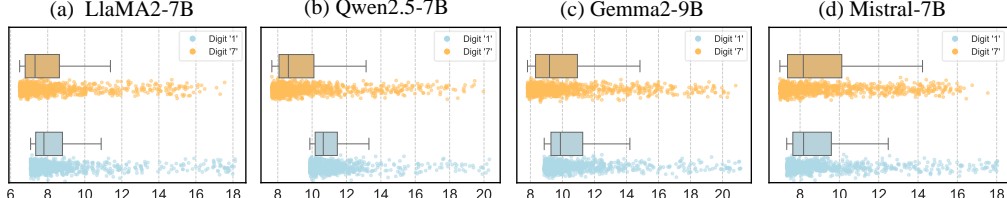

Figure 7: FFN neuron-level selectivity distributions for digit '1' (blue) and digit '7' (orange) in four open-source LLMs: (a) LLaMA2-7B, (b) Qwen2.5-7B, (c) Gemma2-9B, and (d) Mistral-7B. For each digit, we independently select the top 1000 neurons with the highest selectivity. Across all models, neurons most selective for digit '1' exhibit higher selectivity scores than those most selective for digit '7', revealing a stronger model-internal bias toward lower digits.

Table 2: Effect of pruning top 0.01% most digit-1-selective neurons. **Corected - Prop.** denotes the proportion of all test samples that were originally incorrect but become correct after pruning. **Original - Prop.** denotes the original frequency of digit 1 in model outputs. **Pruned - Prop.** denotes the digit 1 frequency after pruning.

| Model | | Multistep Reasoning Tasks | | | |
|---|---|---|---|---|---|
| | | Evaluate | Linear_1d | Nearest Integer root | Sequence Next term |
| Llama2-7B | Coreccted - Prop. | 1.36 % | 0.76 % | 0.19 % | 0.18 % |
| | Original - Prop. | 16.26 % | 16.21 % | 11.91 % | 11.70 % |
| | Pruned - Prop. | 11.17 % | 13.10 % | 6.6 % | 9.74 % |
| Mistral-7B | Coreccted - Prop. | 1.22 % | 1.94 % | 0.35 % | 0.54 % |
| | Original - Prop. | 15.63 % | 14.49 % | 21.64 % | 11.90 % |
| | Pruned - Prop. | 11.85 % | 10.92 % | 10.71 % | 10.61 % |
| Qwen2.5-7B | Coreccted - Prop. | 3.49 % | 5.05 % | 4.96 % | 4.73 % |
| | Original - Prop. | 16.45 % | 15.85 % | 16.25 % | 14.06 % |
| | Pruned - Prop. | 14.72 % | 13.86 % | 13.13 % | 12.29 % |
| Gemma2-9B | Coreccted - Prop. | 1.69 % | 2.14 % | 1.94 % | 1.16 % |
| | Original - Prop. | 13.66 % | 11.04 % | 17.49 % | 11.36 % |
| | Pruned - Prop. | 12.53 % | 10.91 % | 14.22 % | 11.23 % |

If successful, such targeted correction would provide concrete evidence of a causal link between digit bias and numerical hallucination.

Table 3: Corrected - Prop. on arithmetic benchmarks after pruning.

| Model | GSM8k | SVAMP |
|---|---|---|
| Llama2-7B | 2.35 % | 1.60 % |
| Mistral-7B | 1.97 % | 1.40 % |
| Qwen2.5-7B | 2.12 % | 0.70 % |
| Gemma2-9B | 0.08 % | 0.50 % |

## 5.1 Locating and Pruning Biased Neurons

Each FFN neuron's output contributes directly to the residual stream and thus influences final token prediction. We estimate a neuron's contribution to digit bias by isolating its standalone output and computing its Digit Selectivity Score toward each digit.

Formally, for the $n$-th neuron in layer $m$, we define its bias score toward digit $i$ as:

$$\mathbf{biscore}_{n,i}^m = \mathbf{DSC}_i\left(\mathrm{f}_{\text{int}}^{m,n} \cdot \mathbf{w}_{\text{out}}^{m,n}\right) \tag{2}$$

Here, $\mathrm{f}_{\text{int}}^{m,n}$ is the neuron's activation, and $\mathbf{w}_{\text{out}}^{m,n}$ is its corresponding output direction. This scalar product represents the neuron's individual contribution to the final output, which can be interpreted in terms of digit selectivity via the DSC. Given that individual neurons may encode multiple features or functions [41], indiscriminate pruning across all decoding steps may disrupt general model behavior. Therefore, we prune only the top 0.01% most digit-1-selective neurons, and activate this intervention only during the generation of digit tokens. Furthermore, to minimize the disruption to the model's reasoning path, the generation token after pruning is restricted to digit token only.

> **Comparing Model Behavior With and Without Pruning**
>
> **Prompt 1:** Solve -2005 = -1453*p + 1977*p + 3235 for p.
> **Generation (no pruning):** ... Simplify the left side: \[-5240 = 524p\] 3. **Solve for $p$** by by dividing both sides by 524: \[p = \frac{-524**1**}{524}\]... ✗
> **Generation (pruning):** ... Simplify the left side: \[-5240 = 524p\] 3. **Solve for $p$** by by dividing both sides by 524: \[p = \frac{-524**0**}{524}\]... ✓
>
> **Prompt 2:** Solve 366*k + 9029 = 14885 for k.
> **Generation (no pruning):** To solve for k in the equation 366k + 9029 = 14885, we can use the following steps: 1.Subtract 9029 from both sides of the equation to get: 366k = 5856 2. Divide both sides of the equation by 366 to get: k = 16**1** Therefore, the value of k is 161. ✗
> **Generation (pruning):** To solve for k in the equation 366k + 9029 = 14885, we can use the following steps: Step 1: Subtract 9029 from both sides of the equation to get: 366k = 5856 Step 2: Divide both sides of the equation by 366 to get: k = 5856 / 366 k = 16 Therefore, the value of k is 16. ✓
>
> **Prompt 3:** Solve -47613*w = -47331*w + 14100 for w.
> **Generation (no pruning):** To solve the equation $-47613w = -47331w + 141**1**0$ for $w$... ✗
> **Generation (pruning):** To solve the equation $-47613w = -47331w + 141**0**0$ for $w$...✓

Figure 8: Examples showing model responses before and after pruning the top 0.01% of neurons biased toward digit 1. In each cases, pruning corrects an originally erroneous sample by rectifying an intermediate step, demonstrating a causal relationship between digit bias and numerical hallucination.

## 5.2 Debiasing Results and Causal Insight

Table 2 reports two key effects of pruning the top 0.01% most digit-1-selective neurons: a notable reduction in the generation frequency of digit 1, and a measurable correction rate among previously erroneous outputs. Manual inspection confirms that most of these corrected outputs originally featured small-digit hallucinations as shown in Figure 8. Consistent with these findings, Table 3 shows that similar correction effects also emerge on real-world mathematical reasoning benchmarks (GSM8k [42] and SVAMP [43]). Given the Complexity of multi-step numerical reasoning, such corrections are unlikely to be coincidental. Rather, these corrections suggest that digit bias plays an active role in driving the model off its correct reasoning path by skewing generation toward over-frequent digits like 1. If digit bias were not a causal factor in hallucination, such targeted suppression would not yield consistent improvements on biased outputs. This result provides empirical support for a causal link: **digit bias is not merely a statistical artifact, but a mechanistic contributor to numerical hallucination**.

**Takeaway.** While our pruning method is coarse and may negatively affect some previously correct predictions, its ability to reliably correct biased errors underscores the causal link between digit bias and hallucination. Rather than being a general-purpose debiasing tool, this method serves as a probing mechanism to deepen our understanding on how low-level statistical priors manifest as systematic reasoning failures in LLMs.

## 6 Conclusions

This work investigates the overlooked role of digit bias in LLMs. We begin by showing that pretraining corpus exhibit a logarithmic digit distribution consistent with Benford's Law. LLMs internalize this skew, as evidenced by their tendency to overgenerate smaller digits during numerical generation. Notably, in incorrect answers, the first erroneous digit often falls among the smaller digits, suggesting that this bias not only shapes output preferences but also contributes to numerical hallucination. Mechanistically, we trace this bias to the final feed-forward layers of LLMs, where a subset of highly digit-selective neurons encode preferences aligned with corpus statistics. Finally, we propose a lightweight neuron pruning strategy that corrects a portion of biased errors, offering causal evidence that fine-grained digit biases can directly cause numerical hallucination. These findings highlight how low-level statistical priors from pretraining data can affect high-level behavior in LLMs.

**Limitations and future work.** While our work reveals a compelling link between digit bias in the pretraining corpus and numerical hallucinations in LLMs, it has several limitations. Most importantly, although we observe a strong correlation between digit frequencies in the training data and biased generation behavior, we do not claim a causal relationship; establishing causality would require controlled interventions during training, which we leave for future work. In addition, our experiments are conducted on relatively small-scale decoder-only LLMs (7B–9B parameters) with standard MLP architectures. Whether similar patterns of bias and internal activation dynamics persist in larger models or those employing Mixture-of-Experts (MoE) remains an open question. Finally, our pruning strategy offers causal insights into digit-selective neurons but is coarse, potentially disrupting correct generations and limiting accuracy gains. We believe that more fine-grained or adaptive debiasing methods could yield stronger performance improvements. However, developing such techniques is beyond the scope of this work and is left for future investigation.

# 7   Acknowledgements

This work is supported by a Start-up Grant from Nanyang Technological University and jointly funded by the Singapore Ministry of Education (MOE) under a Tier-1 research grant.

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

# A  Additional Experiment Results

## A.1  Result on Digit Bias Benchmark of across Models

We plot the digit distribution in generated outputs across all models on the Digit Bias Benchmark as shwon in Figure 9. All models show significant overgeneration of small digits, with digit 1 being especially dominant. This consistent trend highlights the generality of digit-level generation bias across open-source LLMs.

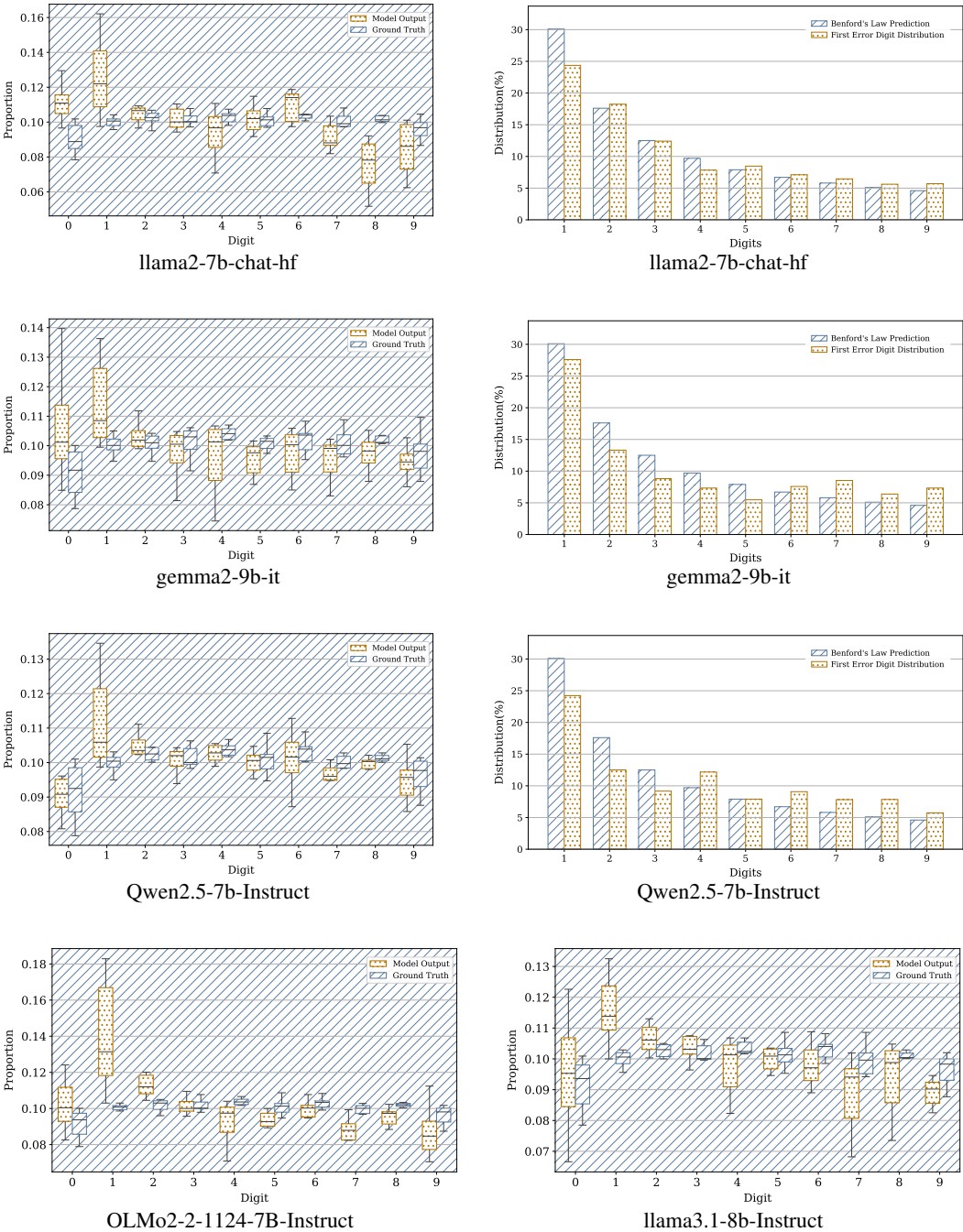

Figure 9: Digit generation bias across models on the Digit Bias Benchmark. Since OLMo and LLaMA3.18B employ multi-digit tokenization schemes, first-error-digit distribution analyses are not applicable and thus omitted for these models.

## A.2 Digit Selectivity

Figure 10 visualizes the selectivity of FFN neurons across all digit tokens. The distributions are clearly skewed, indicating that more neurons are specialized toward frequent digits like '1', suggesting an uneven allocation of model capacity that may underlie observed generation biases.

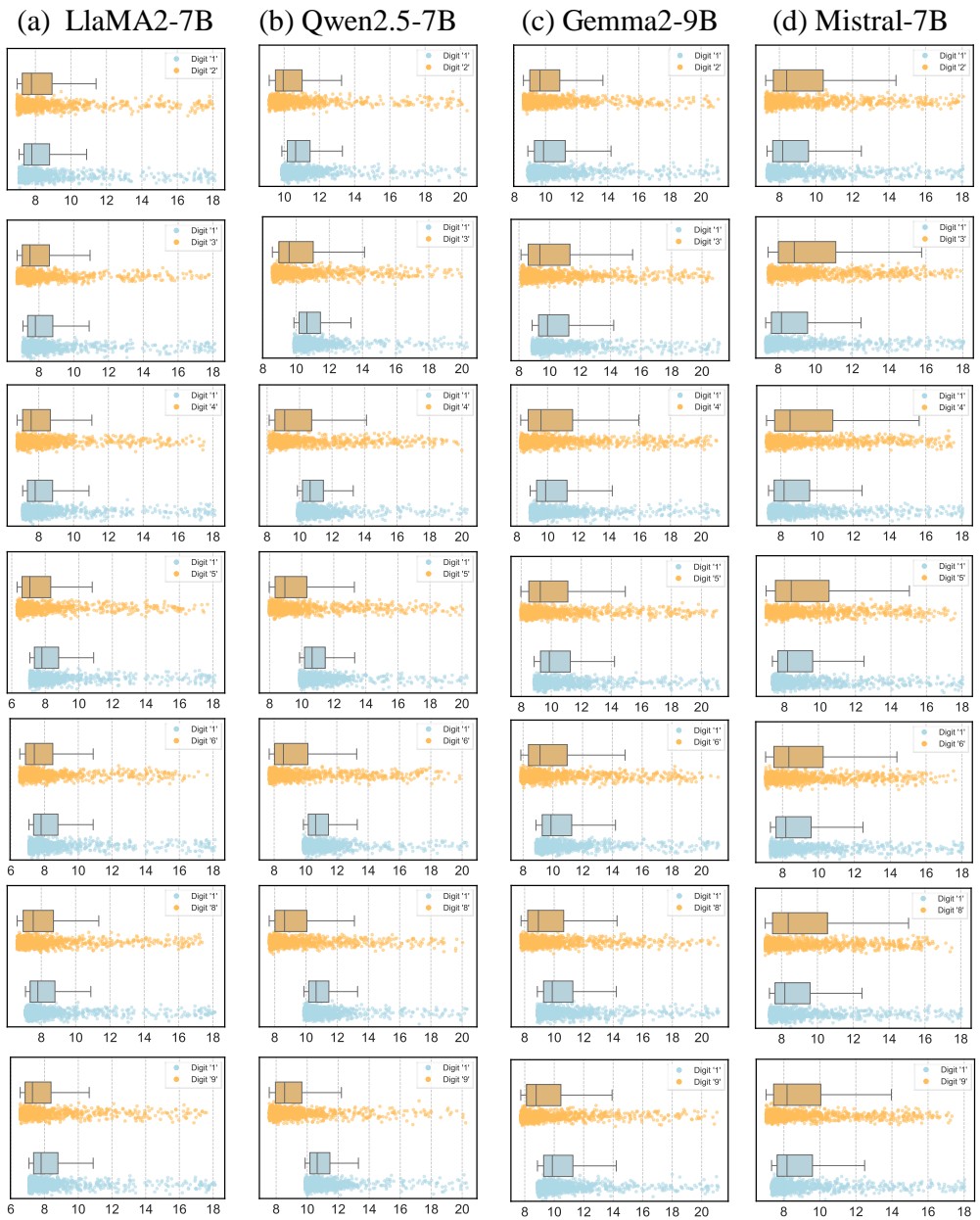

Figure 10: FFN neuron-level selectivity distributions for digit '1' (blue) versus all other digits (orange) across four open-source LLMs: (a) LLaMA2-7B, (b) Qwen2.5-7B, (c) Gemma2-9B, and (d) Mistral-7B. For each digit, the top 1000 neurons with the highest selectivity scores are selected independently.

## A.3 Additional Pruning Examples and Results

Figure 11 presents additional examples of samples corrected after pruning. Many of the original errors occur in surprisingly simple scenarios, such as repeated or trivial arithmetic, highlighting how digit bias can subtly but significantly affect digit generation and lead to failure even in straightforward cases.

---

**Comparing Model Behavior With and Without Pruning**

**Prompt 1:** Solve 0 = -90*w – 12 – 708 for w.
**Generation (no pruning):** To solve for $w$, we need to isolate the variable $w$ on one side of the equation. First, we can simplify the equation by combining like terms: $$0 = -90w - 12 - 708$$. $$0 = -102w - 720$$. Next, we can divide both sides of the equation by $-102$ to get: $$w = -720/102$$. ✗
**Generation (pruning):** To solve for $w$, we need to isolate the variable $w$ on one side of the equation. First, we can simplify the equation by combining like terms: $$0 = -90w - 12 - 708$$. $$0 = -90w - 720$$. Next, we can divide both sides of the equation by $-90$ to get: $$w = frac{720}{90}$$. ✓

**Prompt 2:** Solve 38*w = 75 - 75 - 1026 for w.
**Generation (no pruning):** Let the equation given is: \[ 38w = 75 - 75 - 1126 \]... ✗
**Generation (pruning):** Let the equation given is: \[ 38w = 75 - 75 - 1026 \]... ✓

**Prompt 3:** Solve 78 + 491 + 817 = 28*x + 266 for x.
**Generation (no pruning):** ... 4. **Perform the division:** \[x = 40 \] So, the solution for $x$ is is 41. ✗
**Generation (pruning):** ... 4. **Perform the division:** \[x = 40 \] So, the solution for $x$ is is 40. ✓

**Prompt 4:** Solve 70 = -1160*m - 6852 - 3518 for m.
**Generation (no pruning):** To solve the equation $71 = -1161m - 6852 - 3518$ for $m$ ... ✗
**Generation (pruning):** To solve the equation $71 = -1160m - 6852 - 3518$ for $m$ ... ✓

**Prompt 5:** Solve 38*a - 1664 = -36*a - 69*a + 39*a for a.
**Generation (no pruning):** ... Simplify the left side: \[ 104a - 1664 = 0\] 3. **Isolate the term with $a$:** Add 1664 to both sides to move the constant to the right side: \[104a = 1164\]... ✗
**Generation (pruning):** ... Simplify the left side: \[ 104a - 1664 = 0\] 3. **Isolate the term with $a$:** Add 1664 to both sides to move the constant to the right side: \[104a = 1664\]...✓

**Prompt 6:** What is the next term in -106527, -106453, -106379?
**Generation (no pruning):** The pattern is adding 104 to each term. * -106527 + 104 = -106453 * -106453 + 104 = -106379 * -106379 + 104 = **-106275** Therefore, the next term is **-106275**. ✗
**Generation (pruning):** The pattern is adding 74 to the previous term. * -106527 + 74 = -106453 * -106453 + 74 = -106379 * -106379 + 74 = **-106305** Therefore, the next term is **-106305**. ✓

---

Figure 11: Examples showing model responses before and after pruning the top 0.01% of neurons biased toward digit 1. In each cases, pruning corrects an originally erroneous sample by rectifying an intermediate step, demonstrating a causal relationship between digit bias and numerical hallucination.

# B Additional Implementation Details

## B.1 Experiment Setup

To ensure the accuracy and reproducibility of all results, we employed greedy decoding for generation. Additionally, to achieve fully accurate statistical outcomes, we utilized the DeepSeek API to extract the answers model generated for each sample, thereby preventing potential omissions in script-based statistics due to variations in the position of answers within the responses.

## B.2 Tokenization Method

LLMs divide numbers into segmented tokens rather than representing the entire number as a single token. Different LLMs employ various tokenization methods, including *one-digit tokenizers* and *multi-digit tokenizers*. Benford's Law states that in many real-life sets of numerical data, the leading digit is likely to be small. In other words, regardless of the type of tokenizer, the distribution of number tokens in pretraining data is likely to be skewed. Therefore, in this paper, we use six models with two different tokenizers to investigate this phenomenon of digit bias as shwon in Figure 12. LLaMA2-7B, Mistral-7B, Qwen2.5-7B, and Gemma2-9B employ single-digit tokenizers, whereas LLaMA3.1-8B and OLMo2-7B utilize multi-digit tokenization schemes.

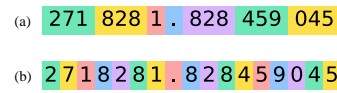

Figure 12: (a) multi-digit tokenizer. (b) single-digit tokenizer.

## B.3 Prompt Templates

We provide the exact prompt templates used for Identification task and Digit Bias Benchmark in table 4 and table 5.

Table 4: Prompt Templates Used in Identification Task

| **Identification Prompt Template** |
| --- |
| 1. What is the result when the last term of the sequence is multiplied by two? [...]
2. What is the outcome when the final term of the sequence is doubled? [...]
3. What is the product of the sequence's last term and two? [...]
4. What is the result of multiplying the sequence's last term by two? [...] |

# C Use of existing assets

## C.1 Models

Table 6: The list of models used in this work.

| Model | Accessed via | License |
| --- | --- | --- |
| Qwen2.5-7B-Instruct | Link | Apache license 2.0 |
| gemma-2-9b-it | Link | Gemma Terms of Use |
| Mistral-7B-Instruct-v0.3 | Link | Apache license 2.0 |
| Llama-3.1-8B-Instruct | Link | Llama 3.1 Community License Agreement |
| Llama-2-7b-chat-hf | Link | Llama 2 Community License Agreement |
| OLMo-2-1124-7B-Instruct | Link | Apache license 2.0 |

Table 5: Prompt Templates Used in Digit Bias Benchmark

| Addition | Division |
|---|---|
| {p} + {q}
{p}+{q}
Work out {p} + {q}.
Add {p} and {q}.
Put together {p} and {q}.
Sum {p} and {q}.
Total of {p} and {q}.
Add together {p} and {q}.
What is {p} plus {q}?
Calculate {p} + {q}.
What is {p} + {q}? | Calculate the division of {q} by {p}.
Divide {q} by {p}.
What is the quotient of {q} divided by {p}?
What is {q} divided by {p}?
Find {q} divided by {p}.
Compute {q} ÷ {p}.
Solve {q} divided by {p}. |
| **Subtraction** | **Multiplication** |
| {p} - {q}
Work out {p} - {q}.
What is {p} minus {q}?
What is {p} take away {q}?
What is {q} less than {p}?
Subtract {q} from {p}.
Calculate {p} - {q}.
What is {p} - {q}? | {p} × {q}
Calculate {p} × {q}.
Work out {p} × {q}.
Multiply {p} and {q}.
Product of {p} and {q}.
What is the product of {p} and {q}?
{p} times {q}
What is {p} times {q}? |

| Evaluate |
|---|
| Let {c(x) = f(x)}.  What is {c(a)}?
Let {c(x) = f(x)}.  Determine {c(a)}.
Let {c(x) = f(x)}.  Give {c(a)}.
Let {c(x) = f(x)}.  Calculate {c(a)}. |
| **Nearest Integer Root** |
| What is the {num-th} root of {p} to the nearest integer?
What is {p} to the power of {1/num-th}, to the nearest integer? |
| **Linear_1d** |
| Solve {eqution} for {r}. |
| **Sequence Next Term** |
| What comes next:  {sequence}?
What is next in {sequence}?
What is the next term in {sequence}? |

## C.2   Dataset

Table 7: The list of datasets used in this work.

| Dataset | Accessed via | License |
|---|---|---|
| olmo-mix-1124 | Link | Open Data Commons License Attribution |
| mathematics_dataset | Link | Apache license 2.0 |

# D   Compute statement

All experiments presented in this paper were run on a cluster of four NVIDIA GeForce RTX 3090 GPUs with 24GB of memory and using a single 24GB memory GPU. Each model requires an average of 50 hours to complete a full run across the entire benchmark.

