# OpenReview forum: "Benford’s Curse: Tracing Digit Bias to Numerical Hallucination in LLMs"
_NeurIPS.cc/2025/Conference — NeurIPS 2025 poster_

### Official Review · Reviewer_TDjj · 2025-06-28

**Clarity:** 4
**Significance:** 2
**Originality:** 3
**Rating:** 5
**Confidence:** 4

**Summary:**

This paper investigates the origin of digit bias in LLMs - the empirical observation that LLMs are more likely to be accurate on mathematical problems when the leading digit is small. The authors link this observation in LLMs to observations from naturalistic text data, where it's been shown empirically that leading digits in written text are more likely to be sampled from smaller numbers (Benford's law).  The authors conjecture that the distribution of digits in pretraining data might as well follow Benford's law and argue that this data bias may contribute to digit bias in LLMs. They also design a benchmark of numerical tasks with even digit distribution to further substantiate the bias effects. Using a combination of existing and novel interpretability techniques they further diagnose the origin of the digit bias in the model components, and propose a neural pruning strategy based on identifying selective neurons to mitigate these bias effects.

**Questions:**

1. Have you conducted any experiments/analyses on any publicly released mathematical benchmarks and do you have any findings that you could share that extend your current results?

2. The digit distribution of pretraining corpora has an interesting trend deviation for digits 7+ (and particularly 9) compared to the trends predicted by Benford's law, almost resembling mild U-shape. Do you have any guesses/ideas why this is the case?

3. Your selected approach to mitigating digit bias involves identifying neurons and pruning them - have you also considered how model post-training could affect/mitigate bias in the pretraining data? If not, can you speculate on whether this could be an effective approach?

4. You highlight the best improvement in Table 2 for Qwen2.5-7B - do you have any idea why this particular model benefited the most from neuron pruning?

5. In numerical cognition a phenomenon of subitizing is well characterized in humans and animals - the rapid, confident and accurate recognition of small numbers (up to 3 or 4). Do you see any reason to believe that there might be such phenomenon in LLMs beyond data statistics?

**Ethical Concerns:**

["NO or VERY MINOR ethics concerns only"]

**Final Justification:**

Concerns have been addressed with additional experimental data.

**Limitations:**

yes

**Paper Formatting Concerns:**

no conerns

**Quality:**

4

**Strengths And Weaknesses:**

### Strengths:
* Well-defined and empirically demonstrated problem of digit bias in LLMs, with extensive and technically sound experiments and analyses to illustrate the bias effects across different types of LLMs, including a controlled benchmark with word math problems ranging in difficulty
* Comprehensive mechanistic interpretability analyses to study how the bias evolves across different layers/modules in the transformer architecture
* Proposed neural pruning method based on aforementioned analyses to improve performance on the proposed benchmark, with positive results on the proposed benchmark

### Weaknesses:

* The paper effectively highlights LLM bias through its analyses and experiments, but its relevance to a broader audience is currently limited due to a lack of demonstrated impact on widely used benchmarks like MATH or GSM8K, or downstream applications such as VQA where numerical reasoning might be important. To significantly broaden its impact, the work should incorporate analyses using established benchmarks and demonstrate improved performance more generally, in its current form this paper might be more of interest to an audience at a targeted workshop

* While positive, the suggested mitigation strategy to deal with digit bias is specifically done only with the digit 1, and the results are variable (ranging from 0.18%-5.05% improvement) and model/task-dependent, suggesting limited significance. Also it involves identifying selective neurons (a method that may be susceptible to changes in parameters), limiting generality that may be present in alternative approaches (e.g. better pretraining data curation). It is unclear how and if this would affect previously correct results

---

> ### Author Rebuttal · Authors · 2025-07-31
>
> We sincerely appreciate the reviewer’s time and effort in reviewing our paper. Your thoughtful comments have been invaluable in helping us improve our work. Below, we provide our responses to each of the issues mentioned.
>
> **W1: The paper offers insightful bias analysis, but its impact is constrained without validation on standard benchmarks or practical applications. (Q1: More mathmetical benchmarks)**
>
> We thank the reviewer for the thoughtful suggestion. As this work is a fundamental study exploring if data bias leads to generation bias and, subsequently, hallucinations in LLMs, our primary focus is on understanding the phenomena, rather than on demonstrating the effectiveness or generalizability of debiasing methods, which denotes a secondary concern at this stage. However, we still agree with the reviewer’s point that evaluating the impact on broader benchmarks would strengthen the paper’s relevance. We have extended our evaluation to include GSM8K and SVAMP. The corrected proportion results are as follows. Qwen and Gemma had already achieved high accuracy on SVAMP, so the improvement was minimal.
> | Benchmark | Llama2-7B | Mistral-7B | Qwen2.5-7B | Gemma2-9B |
> |:---------:|:---------:|:----------:|:----------:|:---------:|
> | GSM8K     |   2.35%   |    1.97%   |    2.12%   |   0.08%   |
> | SVAMP     |    1.6%   |    1.4%    |    0.7%    |    0.5%   |
>
> ---
> **W2: The mitigation strategy is limited to digit 1, with modest and task-dependent improvements. The method's reliance on selective neurons may hinder generalizability, and its impact on previously correct results remains unclear.**
>
> We would like to discuss them one by one to clarify our methodology, the interpretation of our results, and the overall contribution of our work.
>
> 1. **On the Strategy Being Limited to Digit '1'.** We acknowledge the roughness of our approach. However, this focus was driven by the main objective of our study: to investigate the origin of digit bias and its potential causal role in numerical hallucination. While the method may appear coarse, it remains simple and effective for validating our central hypothesis. Also, the intervention is technically not limited to digit ‘1’. In fact, we have tested the cosine similarity between digits in the embedding space and found that the closer two digits are, the higher their similarity (for example, in LLaMA2, the cosine similarities for digit 1 with 0-9 digits are: 0.68, 1, 0.78, 0.72, 0.68, 0.68, 0.65, 0.64, 0.62, 0.62). Furthermore, we observed that a neuron's preference score towards each digit follows a unimodal distribution, meaning that when we prune neurons for digit '1', we are effectively intervening in the direction of smaller digits, not just digit '1'. Therefore, while our current focus is on digit '1', the underlying mechanism we address extends beyond this digit and has broader implications for smaller digits in general.
>
> 2. **On the Variable Results and Their Significance.** The observed variability across tasks and model is likely due to the different difficulty levels and the performance characteristics of the models. For instance, in the Nearest Integer Root task with LLaMA2, pruning resulted in a substantial improvement—from answering only two questions correctly to answering six. While this demonstrates a significant relative gain, in this larger-scale task with over two thousand problems, the improvement was only 0.18%. This suggests that the results’ variability is more related to the nature of the tasks and the model's performance on them, rather than the effectiveness of the pruning approach itself.
>
> 3. **On the Reliance on Selective Neurons and Generalizability.** We appreciate and agree with the reviewer’s insightful comment. However, in this paper, based on the analyses conducted in Section 4, we believe that identifying selective neurons is a natural and effective method to validate the causal link between digit bias and the imbalanced distribution of selective neurons. This aligns closely with our original research goal, which was to explore the causal mechanism behind digit bias.
>
> 4. **On the Unclear Impact on Previously Correct Results.** When pruning is applied excessively, there is a risk of replacing small digits that are used as step numbers or sequence identifiers (e.g., the digit '1' being replaced by newline characters such as \n\n). This could lead to unintended errors in the final generated content, affecting the model’s overall accuracy and output.
>
> ---
> **Q2: The digit distribution for digits 7 and above (especially 9) in the pretraining corpus shows a U-shape pattern, deviating from Benford's law. What could explain this trend?**
>
> We hypothesize this is not due to a single factor, but rather a two-fold effect stemming from the composition of our source data and the selection biases.
>
> - Intrinsic Properties of Web and Technical Data: olmo-mix-1124 pre-training dataset is a heterogeneous mixture of diverse data domains. It is primarily composed of web pages from the DCLM-Baseline corpus [1], but is also supplemented with specialized corpora for code and mathematics. These source domains contain inherent numerical patterns—such as years (e.g., 1990s) and commercial prices (e.g., ending in ".99")—that naturally deviate from a pure Benford's Law distribution. This composition provides the "raw material" for the observed over-representation of high digits.
>
> - Pretraining Corpus Processing: The largest component of olmo-mix-1124, DCLM-Baseline, underwent a rigorous selection process using an algorithm to assign a "quality score" to a large pool of web documents. Only the top 10% with the highest scores were retained, focusing on structured, information-dense content like tutorials and factual articles. This filtering process likely amplifies numerical patterns, particularly high digits, which might be common in such documents, contributing to the mild U-shaped curve.
>
> ---
> **Q3: Has post-training been considered as a potential method to mitigate biases in the pretraining data?**
>
> We do agree that post-training approaches, such as fine-tuning on a debiased dataset or employing adversarial training to specifically target digit bias, could help mitigate this issue effectively.
>
> Despite this, we ultimately chose pruning as our approach for two key reasons:
> - Causal Evidence: Pruning allowed us to provide direct causal evidence linking digit bias to model behavior, which was central to our study.
>
> - Resource Constraints: Due to limited computational resources (only four RTX 3090 GPUs), conducting post-training or large-scale data curation to pretraining was not feasible within the scope of this work.
>
> We believe that exploring data curation methods in future studies, in combination with post-training strategies, could further improve the model's ability to mitigate biases.
>
> ---
> **Q4: Why Qwen benefited the most from neuron pruning?**
>
> We think this is because Qwen exhibits a more extreme FFN neuron-level digit selectivity bias compared to the other models. As the comparison in Figure 7 illustrates, the separation between the selectivity score distributions for digit '1' and digit '7' is visibly more pronounced in Qwen2.5-7B (Figure 7b) than in the other tested models (Figures 7a, 7c, and 7d). This suggests its internal digit bias is stronger and more concentrated.
>
> Plausibly, this stronger bias leads the model to produce more errors driven by biased generation, including very low-level mistakes such as miscopying the digits of the problem itself. For instance, when prompted with "Let s(b) = -149\*b - 1467. Calculate s(**-10**).", the model mistakenly answered: "To calculate  s(**-11**)  for the function ( s(b) = -149b - 1467 )". Therefore, the debiasing was most impactful on Qwen precisely because it had the most severe digit bias problem, which manifested all the way from the neuron-level to the final generated content.
>
> ---
> **Q5: Is there any reason that subitizing exists in LLMs beyond data statistics?**
>
> We appreciate the inspiring question. If we define subitizing simply as the ability to quickly recognize small quantities without actually counting them, we believe it’s possible that LLMs could exhibit a similar behavior, and here’s why:
>
> We have done the experiments before: when we provided the LLM with a numerical sequence and asked it to identify the length of the sequence, models like Mistral and Qwen typically did not count the numbers but gave the answer directly. We found that when the sequence length was below 16, the models usually gave correct answers, but when the sequence length increased up to 128, the models often provided an approximate but incorrect answer. This behavior is strikingly similar to the human ability to subitize small numbers and estimate larger quantities.
>
> However, it is hard for us to envision how this ability could be directly encoded in the training data or emerge through any other means. A similar study [2] has observed cognitive phenomena such as the SNARC effect and the Distance effect in LLMs, but explaining how these abilities emerge remains a challenge.
>
> From a cognitive perspective, subitizing is widely regarded as an innate ability in humans, appearing in infants just a few weeks old [3]. Unlike other cognitive phenomena in LLMs, which are generally influenced by later developmental factors, subitizing is thought to be an inborn skill. Therefore, it is unlikely that LLMs would naturally exhibit this characteristic.
>
> We believe that further research and experiments are needed to explore whether LLMs possess specialized mechanisms that allow for such rapid and accurate recognition of small quantities.
>
>  [1] DataComp-LM: In search of the next generation of training sets for language models
>  [2] Cognitive Effects in Large Language Models, ECAI 2023
>  [3] Are Subitizing and Counting Implemented as Separate or Functionally Overlapping Processes?, NeuroImage Volume 15, Issue 2, February 2002, Pages 435-446

---

> > ### Comment · Reviewer_TDjj · 2025-08-01
> >
> > I thank the authors for the detailed and thoughtful response. Given that my concerns (as well as concerns of some of other authors) have been adequately addressed, I am happy to increase the score. Particularly, the authors have demonstrated that this approach indeed improves performance on more widely used benchmarks for several classes of models thus demonstrating generality. While the performance boost is relatively modest (2.35%, 1.97%, 2.12%) it solidly supports the claims made in the paper and authors show that the performance on other benchmarks is not negatively affected.

---

> > > ### Author Response · Authors · 2025-08-02
> > >
> > > Thank you again for the valuable suggestions and acknowledgment!

---

### Official Review · Reviewer_YkuC · 2025-07-02

**Clarity:** 3
**Significance:** 2
**Originality:** 3
**Rating:** 5
**Confidence:** 2

**Summary:**

Tracing Digit Bias to Numerical Hallucination in LLMs examines why language models hallucinate numbers. Rooted in Benford’s Law, the authors reveal that LLMs overgenerate smaller digits—even with uniform targets—and trace this bias to training data and FFN modules. They introduce a "neuron pruning" fix and demonstrate its effectiveness.

**Questions:**

While the paper introduces a useful quantification of digit bias via Benford’s Law, the core observation that “training data bias leads to hallucinated numeric bias” is not fundamentally new. (and authors acknowledged this point)

Human brain also learns from biased inputs but typically develops internal models that avoid hallucination—this points toward architectural issues in LLMs rather than training data alone. The proposed FFN pruning strategy, while empirically effective in this narrow context, feels ad hoc and may not generalize to broader settings of numeric reasoning or language generation.

Does pruning affect non-numerical reasoning? Would there be any structural mitigations?

**Ethical Concerns:**

["NO or VERY MINOR ethics concerns only"]

**Final Justification:**

The authors have provided additional clarification on the generalizability of the proposed method, which is an important aspect of the work. Furthermore, their justification for adopting a data-centric approach rather than a structural solution is reasonable and well-argued.

**Limitations:**

yes

**Quality:**

3

**Strengths And Weaknesses:**

Strengths

Problem definition and solutions are clearly stated.
Thorough analysis tracing bias source to FFN layers via logit lens and neuron selectivity.
Offers a lightweight intervention (pruning few neurons) to correct bias experimentally.

Weaknesses

Focused *only* on digit bias; does not link findings to broader numeric reasoning or real-world LLM applications.

The mitigation strategy (pruning neurons) lacks evaluation on whether it affects other capabilities.

---

> ### Author Rebuttal · Authors · 2025-07-31
>
> We sincerely appreciate the reviewer’s thoughtful evaluation and valuable feedback on our work. Below, we aim to address the specific concerns and suggestions raised.
>
> ---
> **W1: Focused only on digit bias; does not link findings to broader numeric reasoning or real-world LLM applications.**
>
> We thank the reviewer for this crucial point regarding the link to broader reasoning and applications. We agree that making this connection explicit is vital. In fact, our proposed Digit Bias Benchmark (Section 3.2) includes seven distinct categories of numerical reasoning tasks, such as arithmetic operations, root approximation, linear equation solving, and sequence prediction. Although we have not yet conducted explicit real-world application evaluations, we believe that the observed digit bias is highly likely to manifest in real-world applications as well. This belief is grounded in the breadth and diversity of tasks covered by our benchmark, which simulates many of the numerical reasoning demands encountered in practical settings. We agree that testing on real-world benchmarks is indeed crucial. Therefore, we provide the corrected proportion results on the following benchmarks, where Qwen and Gemma show minimal improvement due to their already high accuracy on SVAMP.
> | Benchmark | Llama2-7B | Mistral-7B | Qwen2.5-7B | Gemma2-9B |
> |:---------:|:---------:|:----------:|:----------:|:---------:|
> | GSM8K     |   2.35%   |    1.97%   |    2.12%   |   0.08%   |
> | SVAMP     |    1.6%   |    1.4%    |    0.7%    |    0.5%   |
>
> ---
> **W2: The mitigation strategy might affect other capabilities in LLMs.**
>
> We thank the reviewer for the thoughtful question. Our pruning strategy is specifically designed to target digit bias, and therefore, we apply it only at the digit generation step, as this is the only stage where mitigation is necessary. This design choice is not merely to avoid unintended side effects, but rather reflects the task-specific nature of the intervention—we do not expect the debiasing mechanism to be relevant outside digit prediction.
>
>
> That said, we acknowledge the importance of assessing broader impacts. To this end, we evaluated the pruned model on two general NLP benchmarks:TruthfulQA and MMLU, and present the results below to demonstrate that overall model performance remains largely unaffected. (To ensure the robustness of our evaluation, we selected the non-numerical tasks anatomy, electrical engineering, formal logic and philosophy from the various MMLU task categories for testing.)
>
> |  Benchmark  |   Llama2-7B   |  Mistral-7B |   Qwen2.5-7B  |   Gemma2-9B   |
> |:-----------:|:-------------:|:-----------:|:-------------:|:-------------:|
> | TruthfulQA  | 23.75 (-0.12) | 48.23 (0.0) | 63.28 (+0.12) | 74.05 (0.0)   |
> | MMLU        | 47.41 (-0.08) | 53.69 (-0.21) | 67.36 (-0.28)   | 66.95 (0.0) |
>
> ---
> **Q1:Training data bias causes hallucinated numeric bias is not fundamentally novel.**
>
> To the best of our knowledge, the most related study (cited in our Related Works) is [1] (referred to as IPTF), which primarily demonstrates that higher term frequencies in the data correlate with better few-shot reasoning accuracy—a frequency-to-accuracy relationship. In contrast, our work focuses on the relationship between frequency and generation bias. The table below highlights the differences between the two papers.
>
> |          Aspect         |                                                           Benford’s Curse                                                          |                                                       IPTF                                                       |
> |:-----------------------:|:----------------------------------------------------------------------------------------------------------------------------------:|:----------------------------------------------------------------------------------------------------------------:|
> | Research Focus          | Investigates digit-level bias in LLMs leading to hallucinated numerical errors, rooted in Benford-like distributions.              | Investigates term-level frequency effects on reasoning accuracy.                                                 |
> | Main Hypothesis         | Pretraining data's digit bias is learned and causes systematic numerical errors in generation.                                     | High-frequency terms in pretraining data lead to higher accuracy in reasoning tasks.                             |
> | Investigated Phenomenon | The focus is on the model's propensity to generate specific single digits (0-9), particularly a preference for lower-value digits. | The focus is on how the frequency of entire numbers (e.g., "24" vs. "23") as operands relates to model accuracy. |
> | Methodology             | - Designed digit-balanced benchmarks to isolate bias. - Analyzed internal digit probabilities via Logit Lens.                      | - Correlational analysis between number frequencies and accuracy. - No internal model probing or intervention.   |
> | Findings                | 1. LLMs prefer smaller digits like in Benford's Law.  2. Bias localizes to a few FFN neurons.                                      | 1. Accuracy is correlated with number frequency. 2. Frequency effects persist across model scales.               |
>
> ---
> **Q2: 1. The digit bias issue might lie in LLM architecture rather than training data and 2. the proposed FFN pruning method feels ad hoc and unlikely to generalize to broader tasks.**
> 1. We thank the reviewer for the thoughtful point. While we agree that architectural improvements could, in principle, mitigate biases more effectively, our focus is on understanding how such biases arise and persist in current state-of-the-art LLMs. Architectural changes are non-trivial to implement, often lack interpretability, and may introduce new trade-offs. In contrast, starting from data allows us to isolate and analyze how distributional skew is internalized, and to identify specific mechanisms—such as a small set of FFN neurons—responsible for bias amplification. This data-driven approach does not preclude architectural solutions. Rather, it provides actionable insights into where and how bias is encoded, offering a foundation for principled architectural design in the future.
>
> 2. As we mentioned in W2, since a single neuron may serve multiple functions, to avoid affecting other tasks, we activate this intervention only during the digit generation step. Therefore, this approach can generalize to a broader range of tasks, as demonstrated by the experiments in W2.
>
> ---
> **Q3: 1. Does pruning affect non-numerical reasoning? 2. Would there be any structural mitigations?**
>
> 1. As illustrated in W2 and Q2.2
> .
> 2. We believe that the observed digit bias phenomenon is largely due to the presence of a significant number of meaningless digits in the pretraining corpus. For example, digits that appear frequently without any meaningful context, such as ISBN numbers, telephone numbers or postal codes, can skew the model's learning process. If the pretraining corpus were composed of high-quality, contextually relevant numeric reasoning digit, we believe that even a non-uniform distribution of digits would not lead to this issue. In fact, [2] demonstrated that selectively ignoring meaningless tokens during pretraining significantly improved the model's mathematical capabilities. We analyzed the tokens excluded in that study and found that a large portion of the ignored tokens were meaningless digits, further supporting our hypothesis. Therefore, one potential structural mitigation strategy is to modify the model architecture to avoid learning from meaningless digits. This could be achieved through architectural improvements, such as attention mechanisms that can dynamically identify and suppress the influence of meaningless digits, or by embedding constraints that prevent such digits from being used in critical tasks like numeric reasoning. These approaches would allow the model to focus more effectively on meaningful numeric patterns while minimizing the impact of noisy or irrelevant digits.
>
> [1] Impact of Pretraining Term Frequencies on Few-Shot Reasoning, EMNLP 2022
> [2] Rho-1: Not All Tokens Are What You Need, Neurips 2024

---

### Official Review · Reviewer_gChx · 2025-07-03

**Clarity:** 3
**Significance:** 3
**Originality:** 2
**Rating:** 4
**Confidence:** 3

**Summary:**

This paper investigates how large language models (LLMs) exhibit systematic biases in digit generation, particularly favoring lower leading digits, in ways that resemble Benford’s Law. By analyzing pretraining corpora and evaluating models on custom benchmarks with uniformly distributed digits, the authors show that this bias is learned during pretraining. Through neuron-level analysis, they identify a small subset of digit-selective neurons responsible for the effect. Pruning these neurons reduces the bias and improves numerical accuracy, offering causal evidence that pretraining statistics can shape symbolic failures in LLMs.

**Questions:**

see weakness

**Ethical Concerns:**

["NO or VERY MINOR ethics concerns only"]

**Final Justification:**

I gave Borderline accept (4) initially and the authors partially addressed my concerns, but I don't think it is worth Accept (5) due to the limited scope.

**Limitations:**

Yes.

**Quality:**

2

**Strengths And Weaknesses:**

Strength

- The paper introduces a novel perspective by linking statistical patterns in pretraining data, such as Benford’s Law, to systematic digit bias in LLM outputs.


- It designs a controlled benchmark with uniformly distributed digits, enabling clear and reliable evaluation of model bias in numerical tasks.


- The use of logit-lens analysis and neuron pruning offers strong causal evidence and practical tools for identifying and mitigating fine-grained model biases.

Weakness

- How do you ensure that the overall distribution of digits (0–9) in the benchmark is truly uniform?


- Please clarify the term “first error digits”; what exactly do the values reported in Table 3 B represent?


- In the logit-lens analysis, are the logits produced by attaching the model’s final prediction head to each intermediate layer?


- The paper claims that digit bias primarily emerges in the final layers, yet the layer-wise digit-preference heat-map in Figure 4b shows strong color intensities for earlier layers as well. How is the conclusion that bias is concentrated only in the last few layers justified?


- For the procedure that traces samples with approximately equal digit probabilities at an intermediate layer, the paper should provide the exact computation and formulas; similar details are needed for how the values in Figure 4b are calculated.


- The proposed pruning strategy is coarse and targets bias toward only digit 1, which harms previously correct predictions and limits the method’s generality.


- The study addresses only digit-frequency bias, overlooking other possible sources such as digit co-occurrence biases and operation biases (e.g., addition, subtraction, multiplication, division).

---

> ### Author Rebuttal · Authors · 2025-07-31
>
> Thank you very much for your time, effort, and thoughtful feedback on our paper. Below, we provide a point-by-point response to the concerns and questions you raised, addressing the weaknesses you identified and offering clarifications where needed.
>
> ---
> **Q1: how to ensure that the digit distribution (0–9) in our benchmark is truly uniform？**
>
> The “uniform digit distribution” refers to the answer set digits (0–9) in our seven benchmark tasks, each constructed from the synthetically generated DeepMind Mathematics dataset. To ensure uniformity, we apply an iterative filtering process that removes overrepresented digits samples until the answer distribution across each task closely approximates uniform. This controlled setup ensures that any observed bias stems from the model, not from the benchmark’s label distribution.
>
> ---
> **Q2: Clarify the term “first error digits”; what exactly do the values reported in Figure 3 B represent?**
>
> The term "first error digit" refers to the first digit in the model's final generated answer that differs from the corresponding digit in the ground-truth answer. For example, if the correct answer to a problem is "72" and the model outputs "12", the "first error digit" would be "1" because it is the first digit in the model's output that deviates from the ground-truth answer. Similarly, if the correct answer is "144" and the model generates "134", the "first error digit" would be "3".
>
> This analysis serves a critical diagnostic purpose: **since the distribution of ground-truth answer digits in our benchmark is explicitly controlled to be uniform, we would expect the distribution of "first error digits"—if errors were purely random—to also approximate a uniform distribution.** However, our results show a clear deviation from uniformity: the "first error digits" are significantly skewed toward smaller digits and closely follow Benford’s Law.  Therefore, Figure 3 B suggests that digit bias not only influences the model’s overall preferences but also distorts its generation trajectory when deviating from the correct answer, implicating it as a potential driver of numerical hallucinations.
>
> ---
> **Q3: Are the logits produced by attaching the model’s final prediction head to each intermediate layer in the logit-lens analysis?**
>
> Yes, that is correct. In the logit-lens analysis, we generate the logits for each intermediate layer by multiplying the model's final prediction head—the unembedding matrix—with that layer's output.
>
> ---
> **Q4: If early layers also show strong color intensities in Figure 4b, how can the paper claim that digit bias mainly arises in the final layers?**
>
> We apologize for the confusion, and we appreciate the opportunity to clarify how to interpret Figure 4b.
>
> Figure 4b shows a layer-wise heatmap of normalized digit probabilities. In earlier layers, the model does not exhibit a strong preference for any specific digit, so the normalized probabilities for each digit are approximately uniform, around 0.1. Therefore, the strong color intensities in early layers of Figure 4b reflect that all digits have similar probabilities—around 0.1—due to the model's uncertainty, not due to a strong preference for specific digits.
>
> **Interpret Figure 4b** : Figure 4b focuses on uncertain digit generation steps, defined as those where the digit probability entropy at layer 26 is greater than 3.0. We observe a crucial asymmetry in how different digits are predicted in these uncertain cases:
> - When the model ultimately generates small digits (e.g., digit '1'), the intermediate layers often do not exhibit a strong preference for that digit. Instead, its probability spikes only in the final few layers—indicating that later layers alone are sufficient to push it over the decision boundary.
>
> - In contrast, when generating large digits (e.g., digit ‘7’), the model needs to build up confidence gradually, often requiring a visible preference for the digit already in the mid-layers.
>
> This behavioral asymmetry suggests that the bias toward small digits like '1' is injected primarily in the final layers. In other words, digit '1' can be “promoted late,” whereas larger digits require sustained evidence from middle layers to be generated at all.
> Thus, while early layers show uniform digit probabilities due to high uncertainty, the actual emergence of generation-relevant digit bias—especially the overgeneration of small digits—is concentrated in the final layers. This is consistent with our logit-lens tracing (Section 4.1) and corroborated by neuron-level pruning experiments in Section 5.
>
> ---
> **Q5: The paper should provide the exact computation and formulas for tracing samples with equal digit probabilities in intermediate layers, as well as details on how the values in Figure 4b are calculated.**
>
> Thank you for your thoughtful suggestion.  We have outlined the specific process below and will include the revision in the new version.
>
> 1.  Figure 4b
>
> To identify uncertain digit generation steps for Figure 4b, we proceed as follows:
>
>
> Let $U \in \mathbb{R}^{v \times d}$ denote the unembedding matrix, where $v$ is the vocabulary size and $d$ is the hidden dimension. Let $h^{(26)} \in \mathbb{R}^{d}$ denote the hidden state at Layer 26 for the generation step. We compute the pre-softmax logits via matrix multiplication:$ z = U \cdot h^{(26)} \in \mathbb{R}^{v} $
>
>
> We apply softmax to obtain the full vocabulary distribution:$ P = \frac{\exp(z_i)}{\sum_{j=1}^v \exp(z_j)} $
>
>
> We extract the probabilities corresponding to digit tokens ${0, 1, \dots, 9}$, and normalize them:$ P^{\text{(digit)}}i = \frac{P_i}{\sum_{j=0}^{9} P_j} \quad \text{for } i \in {0, \dots, 9} $
>
>
> Finally, we compute the entropy of this digit distribution: $ H = -\sum_{i=0}^{9} P^{\text{(digit)}}_i \cdot \log P^{\text{(digit)}}_i $
>
>
> We define a digit generation step as an uncertain sample if $H \geq 3.0$, and include it in the analysis for Figure 4b.
>
>
> For each digit, we identify the uncertain samples where the model outputs that digit, apply logit lens decoding at each layer for those steps, and compute the average predicted probability of that digit. These digit-specific averages across layers form the curves shown in Figure 4b.
>
>
> 2. Figure 5
>
> Based on the same logit lens approach described above, we select samples in which the normalized probability $P^{\text{(digit)}}_i$ of the generated digit at layer 17 satisfies $|P^{\text{(digit)}}-x|<=0.01$, with $x$=0.12 at layer 17 (Figure 5a), $x$=0.15 at layer 22 (Figure 5b), and $x$=0.20 at layer 25 (Figure 5c). We then plot how these probabilities evolve across subsequent layers.
>
> ---
> **Q6:The proposed pruning strategy is coarse and targets bias toward only digit 1, which harms previously correct predictions and limits the method’s generality.**
>
> We acknowledge the reviewer's concern about the specificity of our pruning strategy and we'd like to take this opportunity to clarify the rationale behind its design.
> The primary purpose of this intervention was to simultaneously investigate two critical causal links:
> - **The Source of the Bias**: Is the imbalanced distribution of digit-selective neurons the direct cause of the digit bias observed in generation?
> - **The Consequence of the Bias**: Does this digit bias, in turn, act as a mechanistic cause of numerical hallucination?
>
> While such a pruning intervention can have unavoidable side effects such as potentially harming previously correct predictions, we believe its primary contribution is not diminished by this. Its value as a highly effective and straightforward way to prove causality remains. By successfully correcting a portion of previously erroneous samples and measurably reducing the output bias toward digit '1', our pruning method provides strong empirical support for our central claim: that digit bias is a mechanistic contributor to numerical hallucination and originates from later layer FFN neurons.
>
> We thank the reviewer for this valuable point, as it highlights a promising direction for future work. We plan to design more fine-grained or adaptive pruning strategies to mitigate such side effects while retaining the debiasing benefits, as noted in our paper's limitations section.
>
> ---
> **Q7: The study addresses only digit-frequency bias, overlooking other possible sources such as digit co-occurrence biases and operation biases.**
>
> We thank the reviewer for raising this insightful point. However, our work aims to take an important first step in understanding the relationship between data bias and model behavior, focusing on a particularly fundamental and measurable case: digit-frequency bias. Indeed, various forms of numerical bias—such as digit co-occurrence patterns or operation-specific tendencies—are potentially interesting and worth further investigation. We hope our study provides a foundation for broader inquiries into how pretrained models internalize and manifest different types of bias.

---

> > ### Comment · Reviewer_gChx · 2025-08-02
> >
> > Thank you for the detailed rebuttal.

---

### Official Review · Reviewer_QLC4 · 2025-07-03

**Clarity:** 4
**Significance:** 4
**Originality:** 4
**Rating:** 5
**Confidence:** 4

**Summary:**

The paper focus on analysing and addressing the issues of numerical representation and generation in LLMs. The phenomenon of digit bias was pointed out by analysing LLMs in extensive evaluation in numerical reasoning tasks. Through logic lens technique, the digit bias mechanism is discovered and the intervention by pruning to debias is then proposed. The experiments are conducted across different LLMs.

**Questions:**

In the author's opinion, will this debiasing approach also improves performance of the LLMs in other tasks that require numerical processing, such as mathematical reasoning task, or tasks that involve time-series?

**Ethical Concerns:**

["NO or VERY MINOR ethics concerns only"]

**Final Justification:**

In the rebuttal phase, my questions and comments in the review have been addressed. I maintain the overall score of 5: Accept.

**Limitations:**

Yes

**Quality:**

4

**Strengths And Weaknesses:**

The paper presented very interesting phenomenon via the lens of Benford's law: lower digit often appeared as leading digit. The analysis in web-collected corpora and the answer from LLMs clearly showed this (Figure 2 to 4). This finding potentially reveals a way to look at digit bias/numerical hallucination in LLMs, which is important for reasoning task that involves numbers. Findings from line 171 to 177 gives insight to the phenomenon. It is further substantiated when intervention can be applied to debias. The research is conducted with very careful and well-designed experiments. Experiment is convincing and showing the effectiveness of the approach.

---

> ### Author Rebuttal · Authors · 2025-07-31
>
> We sincerely thank the reviewer's acknowledgement of our work. We are greatly encouraged by your positive assessment of our work. Below, we provide our thought to the raised question.
>
> **Q1:Will this debiasing approach also improve performance of the LLMs in other tasks that require numerical processing, such as mathematical reasoning tasks, or tasks that involve time-series?**
>
> Thank you for your insightful question. We believe our method is indeed generalizable, and our benchmark was designed to test precisely this.
>
> Our approach counteracts the phenomenon of "digit bias," which our research shows is a widespread issue in LLMs. To validate this, we constructed the **Digit Bias Benchmark**, a suite of seven diverse tasks that explicitly cover the areas you mentioned. Six of our tasks (e.g., 'Add or Sub', 'Evaluate', 'Linear_ld') are forms of mathematical reasoning, while the 'Sequence Next Term' task is a direct analogue for time-series analysis.
>
> Our evaluations showed that the digit bias is consistently present across these tasks. Crucially, our intervention, which targets the root cause of this bias—specific digit-selective neurons in the model's deeper layers—proved effective. As shown in Table 2, this method successfully corrected a portion of previously erroneous samples across these varied tasks.
>
> Therefore, because our method targets a fundamental, task-agnostic cause and has already demonstrated its effectiveness on a benchmark representative of both mathematical reasoning and time-series tasks, we expect its success to generalize broadly.
>
> We agree that testing on standard, real-world benchmarks is crucial to validate our approach. Therefore, we extended our evaluation to public benchmarks GSM8K and SVAMP.
>
> The results confirm that our method achieves a positive correction rate on these tasks as well. It is worth noting that on the SVAMP benchmark, the improvements for models like Qwen and Gemma were modest. We attribute this to their already high baseline accuracy on this dataset, which left minimal room for error correction. Nevertheless, the method's effectiveness on other models and tasks demonstrates its broader potential.
> | Benchmark | Llama2-7B | Mistral-7B | Qwen2.5-7B | Gemma2-9B |
> |:---------:|:---------:|:----------:|:----------:|:---------:|
> | GSM8K     |   2.35%   |    1.97%   |    2.12%   |   0.08%   |
> | SVAMP     |    1.6%   |    1.4%    |    0.7%    |    0.5%   |

---

> ### Comment · Reviewer_QLC4 · 2025-08-05
> **Response to Authors**
>
> Dear Authors,
>
> Thank you for your effort in detail responses and discussions about the phenomenon, the proposed benchmark and the approach. The experiments in GSM8K and SVAMP demonstrate that the proposed approach can also improve performance of models on mathematical reasoning task. The findings about the digit bias is fundamental and the generality of the approach has been empirically showed across tasks. The response adequately addressed my questions and comments in the review.
>
> Kind regards

---

### Decision · Program_Chairs · 2025-09-17

**Decision:**

Accept (poster)

**Comment:**

This paper shows that LLMs exhibit systematic digit bias following Benford's Law patterns from pretraining, in particular demonstrating causal relationship between training data statistics and numerical hallucinations through logit-lens analysis and neuron-level interventions. The work provides technical rigor in connecting corpus statistics to model failures. The authors are encouraged to address generalizability issues as well as the intervention scope in the camera-ready version.